# Pioneer Use of Antimalarial Transdermal Combination Therapy in Rodent Malaria Model

**DOI:** 10.3390/pathogens12030398

**Published:** 2023-03-01

**Authors:** Nagwa S. M. Aly, Hiroaki Matsumori, Thi Quyen Dinh, Akira Sato, Shin-Ichi Miyoshi, Kyung-Soo Chang, Hak Sun Yu, Duc Tuan Cao, Hye-Sook Kim

**Affiliations:** 1Division of International Infectious Disease Control, Graduate School of Medicine, Dentistry and Pharmaceutical Sciences, Okayama University, Okayama City 700-8530, Okayama, Japan; 2Department of Parasitology, Faculty of Medicine, Benha University, Benha 13511, Egypt; 3Department of Biochemistry, Faculty of Pharmaceutical Sciences, Tokyo University of Science, Noda 278-8530, Chiba, Japan; 4Department of Sanitary Microbiology, Faculty of Pharmaceutical Sciences, Okayama University, Tsushima-Naka, Kita-Ku, Okayama City 700-8530, Okayama, Japan; 5Department of Clinical Laboratory Science, College of Health Sciences, Catholic University of Pusan, Busan 46252, Republic of Korea; 6Department of Parasitology and Tropical Medicine, School of Medicine, Pusan National University, Yangsan-si 626-870, Republic of Korea; 7Department of Pharmaceutical Chemistry and Quality Control, Faculty of Pharmacy, Hai Phong University of Medicine and Pharmacy, Hai phong, Vietnam

**Keywords:** transdermal N-89, mefloquine, pyrimethamine, antimalarials, combination, in vivo

## Abstract

We have previously reported 1,2,6,7-tetraoxaspiro [7.11]nonadecane (N-89) as a promising antimalarial compound. In this study, we evaluated the effect of transdermal therapy (tdt) of N-89 in combination (tdct) with other antimalarials as an application for children. We prepared ointment formulas containing N-89 plus another antimalarial drug, specifically, mefloquine, pyrimethamine, or chloroquine. In a 4-day suppressive test, the ED_50_ values for N-89 alone or combined with either mefloquine, pyrimethamine, or chloroquine were 18, 3, 0.1, and 3 mg/kg, respectively. Interaction assays revealed that N-89 combination therapy showed a synergistic effect with mefloquine and pyrimethamine, but chloroquine provoked an antagonistic effect. Antimalarial activity and cure effect were compared for single-drug application and combination therapy. Low doses of tdct N-89 (35 mg/kg) combined with mefloquine (4 mg/kg) or pyrimethamine (1 mg/kg) gave an antimalarial effect but not a cure effect. In contrast, with high doses of N-89 (60 mg/kg) combined with mefloquine (8 mg/kg) or pyrimethamine (1 mg/kg), parasites disappeared on day 4 of treatment, and mice were completely cured without any parasite recurrence. Our results indicated that transdermal N-89 with mefloquine and pyrimethamine provides a promising antimalarial form for application to children.

## 1. Introduction

Malaria is a tropical parasitic infectious disease that infects 241 million people annually and causes 627,000 deaths, about 77% of which are children [1]. The use of artemisinin combination therapy (ACT) as the first-line treatment of malaria was officially recommended by the WHO [2]. Unfortunately, ACT-resistant *P. falciparum* has emerged, and its prevalence is expanding [3,4]. Therefore, there is an urgent need to develop antimalarial drugs with novel structures and to target different metabolic pathways to overcome ACT and other drug-resistant *P. falciparum*.

We have previously studied new antimalarial drug development using several thousand compounds [5,6,7,8]. From these studies 1,2,6,7-tetraoxaspiro [7.11]nonadecane (N-89) was selected as a novel antimalarial candidate. N-89 showed high antimalarial activity in vitro and in vivo as an oral drug [9,10,11]. The characteristic features of N-89 are a small molecular weight (MW = 273), water insolubility, and a ClogP value of approximately 4.5. Although N-89 shows a high antimalarial activity as a single drug, it is desirable to evaluate it in combination with existing antimalarial drugs.

Many of the victims of malaria are children under the age of five and living in areas in which malaria is endemic [1]. We changed the drug form from oral administration to a transdermal type (td) with the goal of easy application to malaria patients, especially children. Td delivery may avoid the shortcomings associated with oral delivery and parenteral injections [12,13,14,15]. The aim of this study was to evaluate the effect of transdermal combination therapy (tdct) of N-89 with other known antimalarials for malaria treatment.

## 2. Materials and Methods

### 2.1. Compounds 

N-89 (Figure 1) was synthesized as described [9]. Polyethylene glycol (PEG) 400 and PEG 4000 were purchased from Nacalai Tesque, Inc. (Kyoto, Japan). Mefloquine, pyrimethamine, and chloroquine were purchased from Sigma Chemical Co. (St. Louis, MO, USA). All other chemicals and reagents were analytical-grade commercial products.

### 2.2. Parasites and Mice

*Plasmodium berghei* (*P. berghei)* NK65 strain parasites were maintained in the laboratory. ICR male mice (The Jackson Laboratory Inc., Kanagawa, Japan) at roughly 5 weeks of age and 27 to 28 g weight were maintained at 25 °C and 55% humidity. The mice were allowed free access to food and water. The study was reviewed and approved by the Ethics Committee of Okayama University. The study was carried out in accordance with the guidelines of the Okayama University Animal Care and Use Committee (approval numbers: OKU-2019307 and OKU-2022933).

### 2.3. Preparation of Transdermal Antimalarial Formula

N-89, mefloquine, and pyrimethamine were prepared as an ointment td formula. PEG400 was used as a solvent with a concentration ratio of PEG400 to PEG4000 8:1 (v:w). Briefly, the ointments were prepared by mixing 20 to 40 mg of N-89 per 1 mL of PEG400, followed by sonication for 1 h at 60 °C. Then, 125 mg of PEG4000 was added to the prepared PEG400 samples in a hot water bath at 60 °C, with mixing on an ointment plate after shaking and cooling at room temperature for 10 min. Mefloquine (3–6 mg/kg) and pyrimethamine (0.1–0.5 mg/kg) were prepared by the same method with no precipitations. Since chloroquine (3 mg/kg) did not dissolve in PEG400, it was dissolved in distilled water first, and then chloroquine solution was added to PEG400 (1% water content). Then, PEG4000 was added to prepare the ointment form. 

### 2.4. Td antimalarial Activity and Combination Interaction Assay In Vivo

The antimalarial effect of td formulations was examined using Peters’ 4-day in vivo test as described [16]. The backs of the mice were shaved to 4 cm^2^ the day before the parasite inoculation. Healthy mice were inoculated with *P. berghei* parasites by iv infusion of 1 × 10^5^ infected erythrocytes. Drug administration on day 0 was performed 2 h after the parasite inoculation. The mice were transdermally administered once a day for 4 consecutive days with N-89 (at doses of 0, 25, 35, and 50 mg/day), mefloquine (at doses of 0, 3, 6, and 8 mg/day), pyrimethamine (at doses of 0, 0.05, 0.1, 0.2, and 0.5 mg/day), and chloroquine (0, 1, 3, and 5 mg/kg). The control group was administered with a vehicle only. Five mice were used for each test point. Thin blood films were made, and the parasitemias were calculated. The results were expressed as the percent reduction in parasitemia compared to the untreated control. ED_50_ and ED_90_ values were calculated from the sigmoid dose–response curve.

The interaction of N-89 with other antimalarials was assessed based on the calculation of the fractional effective dose (FED) of the two compounds (A and B). FED was calculated for each association by dividing the ED_50_ of the drug in combination by the ED_50_ of the drug alone. The sum of these two FED (∑FED) was used to plot isobologram curves. A concave curve (∑FED < 1) indicates a synergistic effect between drug A and drug B, a straight diagonal line (∑FED = 1) indicates an additive effect, and a convex curve (∑FED > 1) indicates antagonism [17,18].

### 2.5. Antimalarial Activity and Cure Effect of Transdermal Combination Therapy 

To evaluate the antimalarial activity and cure effect of td formulations, the backs of the mice were shaved to 4 cm^2^ the day before td drug application. Healthy mice were inoculated by iv infusion of 1 × 10^6^
*P. berghei*-infected erythrocytes. Mice with about 0.2% parasitemia on day 1 were td treated once daily for 4 days with low doses or high doses of antimalarial compounds alone or in combination. Low given doses were N-89 (35 mg/kg), mefloquine (4 mg/kg), or pyrimethamine (1 mg/kg). High doses given were N-89 (60 mg/kg), mefloquine (8 mg/kg), or pyrimethamine (1 mg/kg). The control group was administered a vehicle only. Thin blood films were made every day for 30 days, and the parasitemias were determined. We also observed the appearance of any changes in the skin such as redness, edema, or skin irritation. 

### 2.6. Statistical Analysis

Data were tabulated and analyzed using the computer program Statistical Package for Social Science (SPSS) version 26. Categorical data were presented as numbers and percentages, and quantitative data were expressed as mean ± standard deviation (SD). A student’s *t*-test was used to compare the mean of two groups of numerical (parametric) data. ANOVA was used to compare more than two groups of numerical (parametric) data. For continuous non-parametric data, the Kruskal–Wallis test and post-hoc test were used to determine intergroup comparison. A *p*-value < 0.05 was considered statistically significant. The parasitemia was calculated as follows: (total number of parasitized erythrocytes/total number of erythrocytes) × 100. ED_50_ and ED_90_ values were calculated from dose–response curve analysis. Mean survival time was calculated as follows: sum of the survival time of all mice in a group/total survival time of mice in that group. Survival curves were used to study the number of surviving mice in treated and control mice.

## 3. Results 

### 3.1. Td Antimalarial Activity and Combination Interaction Assay in Vivo

We determined the ED_50_ and ED_90_ values of tested compounds using the 4-day suppressive test. The ED_50_ values for N-89, mefloquine, pyrimethamine, and chloroquine were 18, 3, 0.1, and 3 mg/kg, respectively. The ED_90_ values were 42, 5.5, 0.4, and 6.4 mg/kg, respectively (Table 1).

The effect of the combination of tested drugs is represented by the isobolograms in Figure 2. Synergy was observed with N-89 in combination with mefloquine and pyrimethamine (Figure 2a,b). For the chloroquine combination, all points showed an antagonistic effect (Figure 2c). From the above results, mefloquine and pyrimethamine, which showed synergistic effects with N-89, could be considered as partner drugs for N-89; thus, the following antimalaria activity and cure effects were tested using mefloquine and pyrimethamine.

### 3.2. Antimalarial Activity and Cure Effect of Combined Td Drug Application

The results of low-dose experiments of single- and combined drug applications are shown in Table 2 and Figure 3. Single-drug applications for N-89 (35 mg/kg) and mefloquine (4 mg/kg) caused a partial decrease in parasitemia on days 4 to 6, then increased anew from day 7 and 8, respectively. All mice died on days 20 and 23, respectively. Pyrimethamine (1 mg/kg) caused a decrease in parasitemia to day 5, which then increased again, and all mice died on day 12. Tdct showed decreased parasitemia on day 3, and parasites disappeared from the blood stream on day 5. Parasitemia increased anew on day 12 for N-89-mefloquine tdct and on day 10 for N-89-pyrimethamine tdct. All tdct demonstrated a significant effect in comparison with single-drug administration at days 4, 7, and 9 post treatment (*p* < 0.05) (Table 2). As shown in Figure 3a, the SD values ranged broadly, and so we illustrated individual mouse parasitemias in Figure 3b. This figure is an enlarged view of the change in every mouse parasitemia from day 10 to day 30, to make it easier to see the change in parasitemia after tdct. Two mice survived to the end of the experiment, but with low parasitemia for N-89-mefloquine tdct. Only one mouse for the N-89-pyrimethamine tdct was completely cured without parasite observation and survived to the end of the experiment. Therefore, we tested the high doses of N-89 and mefloquine, but we fixed the pyrimethamine dose to 1 mg/kg, because it showed a cure effect with this dose.

The results of the high dose of td N-89 (60 mg/kg) are shown in Table 3 and Figure 4, either applied alone or combined with mefloquine (8 mg/kg) or pyrimethamine (1 mg/kg). Application of N-89 alone caused a decrease in parasitemia to day 6, then parasitemia increased anew on day 7 to day 14, and all mice died on day 14. In the mefloquine-treated group, parasitemia decreased from day 3, and parasites disappeared from the mice blood stream on days 7 and 8. The parasites were observed again from day 9, and all mice died on day 17. In the pyrimethamine-treated group, parasitemia decreased on day 2, then increased again from day 3, until all mice died on day 16. In the group of combined administration of N-89 with mefloquine, parasitemias partially increased on day 2 and then decreased to zero on day 4 to day 30. The group of combined administration of N-89 with pyrimethamine showed decreased parasitemias from day 2, and the parasite was not observed in the blood from day 4 to day 30 (Figure 4a). The absence of parasites in both combination therapies continued to apparent complete recovery without recurrence (Figure 4c). Detailed changes in parasitemias between day 1 and 9 are shown in Figure 4b. We found no statistical difference in the activity between N-89, mefloquine, and pyrimethamine to day 4. From day 9, there was a statistically significant reduction in parasitemia of combined groups in comparison to single-drug application groups. No statistical difference was observed between the N-89-mefloquine and N-89-pyrimethamine combination groups (Table 3). 

Mice survival time was determined by observation for 30 days. The combination of a low dose of N-89 (35 mg/kg) with mefloquine (4 mg/kg) or pyrimethamine (1 mg/kg) did not cause a complete cure (Table 4 and Figure 3b). In contrast, the combination of a high dose of N-89 (60 mg/kg) with mefloquine (8 mg/kg) or pyrimethamine (1 mg/kg) caused a complete cure, and all mice survived, with survival more than two times the length of the single-drug application groups (Table 4 and Figure 4b). 

## 4. Discussion

Many new antimalarial combination therapy studies have been reported towards the goal of overcoming ACT-resistant *P. falciparum*, including derivatives of artemisinin with other compounds, but despite these efforts, the resistance of *P. falciparum* to ACT remains increasing in prevalence [18,19]. Antimalarial drug development studies for children using artemisinin derivatives were also reported [20,21].

Td applications are a desirable form of drug delivery because of their advantages over other routes of delivery, such as the provision of convenient and pain-free self-administration for patients, ease of use, the ability for multiple dosing, and on-demand delivery of drugs. A transdermal ointment is generally inexpensive, directly enters the systemic circulation, and is less affected by the first pass through the liver [22]. Other non-oral forms of antimalarial drugs for children were reported, such as intramuscular injections and rectal suppositories [23,24].

N-89, a novel compound discovered in our laboratory [9], is thought to contribute to malaria control because of its high antimalarial activity. Mefloquine, pyrimethamine, and chloroquine—which have long half-lives and different basic structures from N-89—were tested as partner drugs for N-89. In the mechanism analysis of N-89, we found the candidate target molecule for *P. falciparum* was endoplasmic reticulum calcium-binding protein (*Pf*ERC) (NCBI accession no. 124803623) [10,11]. *Pf*ERC is essential for parasite growth, and its inhibition causes parasite death. Although the mechanism of action of mefloquine, which has a longer half-life than that of N-89, has not been fully elucidated, it accumulates in lysosomes and forms a complex with heme, leading to protein binding of heme and conversion to hemozoin. Alternatively, it might act on the ribosomal protein in *P. falciparum,* which inhibits parasite protein synthesis [25]. Pyrimethamine inhibits folic acid synthesis by inhibiting parasite dihydrofolate reductase [26].

PEG are synthetic polymers of condensed ethylene oxide and water [27]. For skin formulations, PEGs are primarily used as safe non-irritant solvents [28,29]. We reported the antimalarial effect of transdermal N-89 in vivo [30]. Transdermal N-89 two times/day for 4 consecutive days was required for the cure effect on *P. berghei* NK 65 strain in ICR mice. To evaluate the antimalarial activity of td types for two-drug combination therapies for application in children, we investigated N-89 combined with partner antimalarial drugs as an ointment formula. When N-89 was used in combination with mefloquine or pyrimethamine, it showed strong antimalarial activity and synergistic effect. The combination with existing antimalarial drugs that have long blood half-lives and different mechanisms of action as partner drugs would enhance drug efficacy. In future studies, we will seek to improve the formulation, for example, by creating a patch or tape type with a smaller application size. 

In conclusion, we report for the first time that tdct once daily for 4 days of N-89 combined with mefloquine or pyrimethamine showed high antimalarial activity in an application area of 4 cm^2^ in mice with 0.2% initial parasitemia. This leads to decreased drug side effects and represents a drug application route that is amenable to children. 

## Figures and Tables

**Figure 1 pathogens-12-00398-f001:**
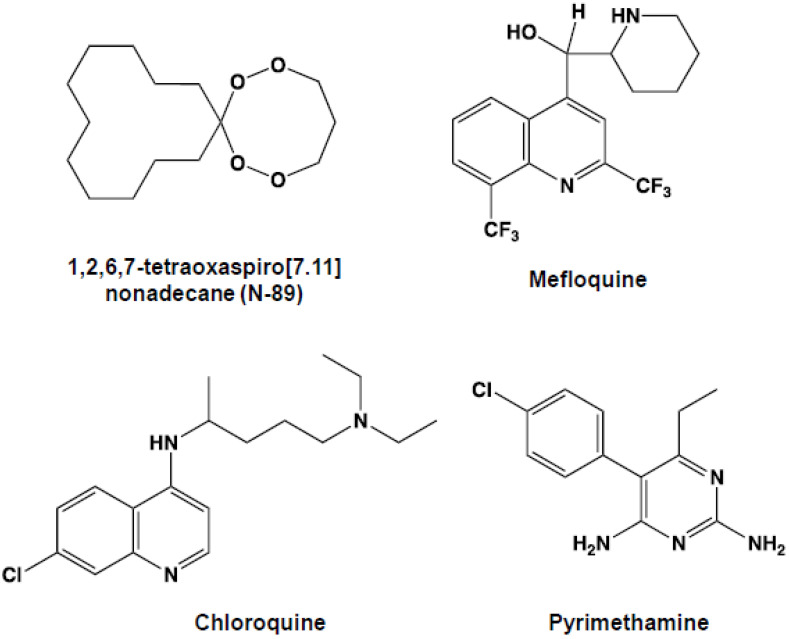
Chemical structures of the tested antimalarial drugs.

**Figure 2 pathogens-12-00398-f002:**
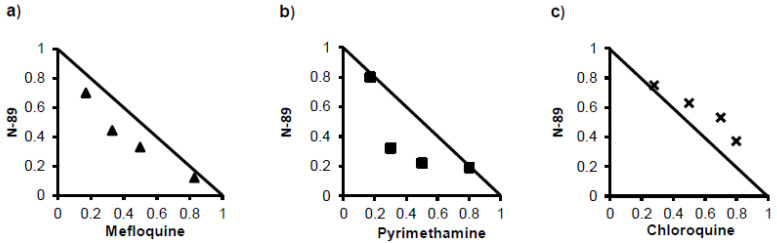
Effects of combination of N-89 with other antimalaria drugs. Isobolograms show the effect of combination of N-89 with mefloquine (**a**), pyrimethamine (**b**), and chloroquine (**c**).

**Figure 3 pathogens-12-00398-f003:**
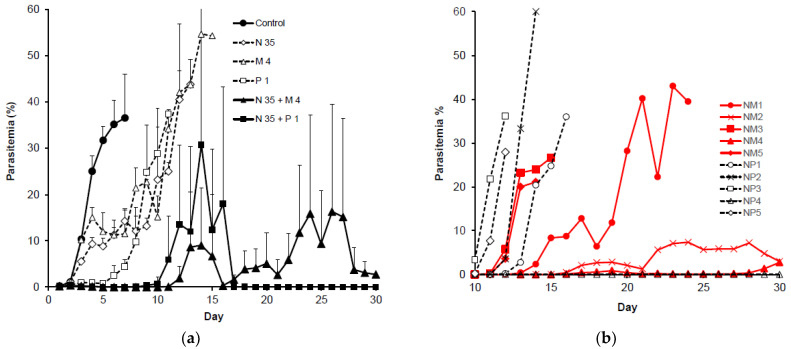
(**a**) Antimalarial activity of tdct (low dose). N 35: N-89 (35 mg/kg), M 4: mefloquine (4 mg/kg), P 1: pyrimethamine (1 mg/kg). Data are expressed as the mean, with a bar showing the SD value. (**b**) Individual mouse parasitemias of tdct. Enlarged view of in vivo antimalarial activity of tdct on days 10 to 30. N: N-89 (35 mg/kg), M: mefloquine (4 mg/kg), P: pyrimethamine (1 mg/kg). NM: N-89 + mefloquine, NP: N-89 + pyrimethamine. (**c**) Survival curve of tdct (low dose). N 35: N-89 (35 mg/kg), M 4: mefloquine (4 mg/kg), P 1: pyrimethamine (1 mg/kg). Survival of the mice was observed for 30 days. The number of surviving mice is plotted against the number of days.

**Figure 4 pathogens-12-00398-f004:**
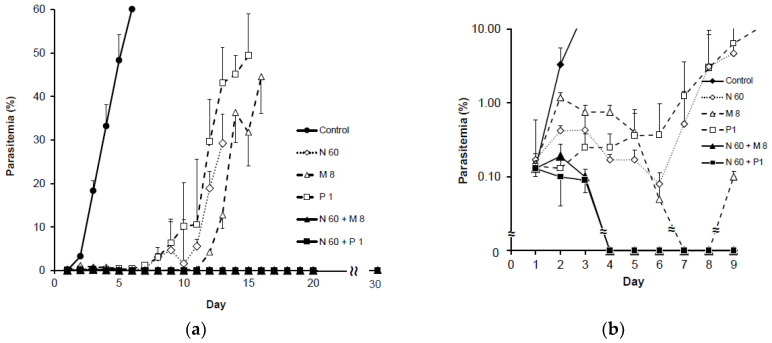
(**a**) Antimalarial activity of tdct (high dose). N 60: N-89 (60 mg/kg), M 8: mefloquine (8 mg/kg), P 1: pyrimethamine (1 mg/kg). Data are expressed as the mean, with a bar showing the SD value. (**b**) Antimalarial activity of tdct between days 0 and 10 (high dose). Enlarged view of in vivo antimalarial activity of tdct for days 0 to 10. N: N-89 (60 mg/kg), M: mefloquine (8 mg/kg), P: pyrimethamine (1 mg/kg). Data are expressed as the mean, with a bar showing the SD value. (**c**) Survival curve of tdct (high dose). N 60: N-89 (60 mg/kg), M 8: mefloquine (8 mg/kg), P 1: pyrimethamine (1 mg/kg). Survival of the mice was observed for 30 days. The numbers of surviving mice are plotted against the number of days.

**Table 1 pathogens-12-00398-t001:** Antimalarial activity of tested compounds by a transdermal route. Data are expressed as mean ± SD value.

Compounds	ED_50_ (mg/kg)	ED_90_ (mg/kg)
N-89	18.0 ± 2.0	42.0 ± 3.2
Mefloquine	3.0 ± 0.7	5.5 ± 0.2
Pyrimethamine	0.1 ± 0.5	0.4 ± 0.1
Chloroquine	3.0 ± 0.3	6.4 ± 0.1

**Table 2 pathogens-12-00398-t002:** Comparison between study groups regarding parasitemias on different days (low dose).

**Parasitemia (%)**						
**Compounds**	**Day 1**	**Day 4**	**Day 7**	**Day 9**	**Day 15**	**Day 30**
Control	0.18 ± 0.02	25.04 ± 3.3	36.58 ± 9.43	-	-	-
N-89	0.23 ± 0.06	9.31 ± 1.36 ^a^	14.29 ± 2.4 ^a^	13.16 ± 1.77	-	-
Mefloquine	0.20 ± 0.03	15.30 ± 2.18 ^a, b^	11.51 ± 5.43 ^a^	22.75 ± 0.59 ^c, d^	54.33	-
Pyrimethamine	0.21 ± 0.04	0.90 ± 0.42 ^a, b, c^	4.39 ± 1.59 ^a, b^	24.81 ± 10.21 ^b^	-	-
N-89 + mefloquine	0.19 ± 0.03	0.12 ± 0.04 ^a, b, c^	0 ^a, b, c, d^	0 ^b, c, d^	6.67 ± 13.30 ^c^	2.74 **
N-89 + pyrimethamine	0.17 ± 0.02	0.10 ± 0.05 ^a, b, c^	0 ^a, b, c, d^	0 ^b, c, d^	12.36 ± 17.47 ^c^	0 ***
*p*-value *	0.2	<0.001	<0.001	<0.001	0.8	

N-89: 35 mg/kg, mefloquine: 4 mg/kg, pyrimethamine: 1 mg/kg. Parasitemia is expressed as mean ± SD value. *: significance < 0.05. a: significance vs. control, b: significance vs. N-89, c: significance vs. mefloquine, d: significance vs. pyrimethamine. **: two mice. ***: one mouse. -: All mice died.

**Table 3 pathogens-12-00398-t003:** Comparison between study groups regarding parasitemias on different days (high dose).

**Parasitemia (%)**						
**Compounds**	**Day 1**	**Day 4**	**Day 7**	**Day 9**	**Day 15**	**Day 30**
Control	0.13 ± 0.01	33.20 ± 4.94	-	-	-	-
N-89	0.17 ± 0.03	0.17 ± 0.03 ^a^	0.52 ± 0.93	4.69 ± 10.02	-	-
Mefloquine	0.15 ± 0.03	0.75 ± 0.19 ^a^	0 ^b^	0.11 ± 0.03 ^b^	31.80 ± 8.37	-
Pyrimethamine	0.14 ± 0.02	0.25 ± 0.13 ^a^	1.24 ± 2.39	6.38 ± 9.98	49.36	-
N-89 + mefloquine	0.13 ± 0.03	0 ^a^	0 ^b, d^	0 ^b, c, d^	0 ^c, d^	0 **
N-89 + pyrimethamine	0.13 ± 0.03	0 ^a^	0 ^b, d^	0 ^b, c, d^	0 ^c, d^	0 **
*p*-value *	0.1	<0.001	<0.001	<0.001	0.001	

N-89: 60 mg/kg, mefloquine: 8 mg/kg, pyrimethamine: 1 mg/kg. Parasitemia is expressed as mean ± SD value. *: Significance <0.05. a: Significance vs. control, b: significance vs. N-89, c: significance vs. mefloquine, d: significance vs. pyrimethamine. **: All experimental mice survived to day 30. -: All mice died.

**Table 4 pathogens-12-00398-t004:** Mean survival time of td single and combined treatment.

**Compounds**	**Survival (Day)** **Low Dose ***	**Survival (Day)** **High Dose ****
Control	6.6 ± 2.3	6.4 ± 2.6
N-89	14.2 ± 2.3	12.8 ± 4.7
Mefloquine	12.2 ± 3.5	15.8 ± 7.2
Pyrimethamine	10.8 ± 4.2	14 ± 5.7
N-89 + mefloquine	23.8 ± 3.0	>30.0 ***
N-89 + pyrimethamine	17.6 ± 5.5	>30.0 ***

*: N-89: 35 mg/kg, mefloquine: 4 mg/kg, pyrimethamine: 1 mg/kg. **: N-89: 60mg/kg, mefloquine: 8 mg/kg, pyrimethamine: 1 mg/kg. Data are expressed as mean ± SD value. ***: All experimental mice survived to day 30.

## Data Availability

The data presented in this study are available on request from the corresponding author (hskim@cc.okayama-u.ac.jp).

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
