# Peer review of "Pioneer Use of Antimalarial Transdermal Combination Therapy in Rodent Malaria Model"

_pathogens, 2023, doi:10.3390/pathogens12030398_

Round 1
Reviewer 1 Report
In the submitted manuscript the authors report for the first time that the transdermal (TD) administration, once daily for 4 days, of N-89 combined with mefloquine or pyrimethamine showed high antimalarial activity in an application area of 4 cm2 in mice with 0.2% initial parasitemia. This leads to decreased drug side effects and represents a drug application route amenable to children. The same group had previously shown that N-89 is a new drug active against malaria parasites in vitro and in vivo.
The TD administration of N-89 together with other antimalarial opens a new and promising field of research. Thus, other antimalarials should be tested by the TD route, which if proven active will greatly facilitate treatment of young children, a quite original idea. The language is clear, the methods are well described and although the figures are somewhat repetitive and complex, they are comprehensible.
TD applications are a desirable form of drug delivery because of their advantages over other routes of delivery, such as a convenient and pain-free self-administration for patients, ease of use, plus the ability for multiple dosing, and on-demand delivery of drugs.
Author Response
We'd like to express our immense appreciation for your time and revision. Thank you so much for trusting our work
Reviewer 2 Report
The article by Hye-Sook Kim et al describes the evaluation of the antimalarial effect of transdermal therapy (tdt) of 1,2,6,7-tetraoxaspiro[7.11]nonadecane (N-89) in combination (tdct) with other antimalarials as an application for children. The ointment formulas containing N-89 plus another antimalarial drug; specifically, mefloquine, pyrimethamine, or chloroquine showed significant activity i.e. as mentioned, in the 4-day suppressive test, the ED50 values for N-89 either alone or in combination with mefloquine, pyrimethamine, or chloroquine, were found to be 18, 3, 0.1, and 3 mg/kg, respectively. The synergistic effect were promising with N-89 and mefloquine/ pyrimethamine. Antimalarial activity and cure effect were compared for single drug application and combination therapy. The data obtained with low doses of tdct [N-89 (35 mg/kg) combined with mefloquine (4 mg/kg) or pyrimethamine (1 mg/kg)] gave only antimalarial effect but not a cure effect. However, high doses [N-89 (60 mg/kg) combined with mefloquine (8 mg/kg) or pyrimethamine (1 mg/kg)] furnished promising result as the parasites disappeared on day 4 of treatment and mice were completely cured without any parasite recurrence. The author finally inferenced that the transdermal N-89 with mefloquine and pyrimethamine could serve as promising antimalarial form for its application to especially children. The antimalarial effect of transdermal therapy (tdt) of 1,2,6,7-tetraoxaspiro[7.11]nonadecane (N-89) is well -studied.
Author Response
We'd like to express our immense appreciation for your time and revision. Thank you so much for trusting our work.
Reviewer 3 Report
Why Nk65 mice are used. Authors should see the effect of drug on Anka strain which more closely mimic the human infection.
I have concerns on how the parasitemia was detected. The parasitemia does not go reach25% at day 4 with this strain. I assume authors have added a very unusual high number of infected RBCs in the mice and that makes their result hard to interpret.
To my opinion based on results shown in Table, it is clear N89 and mefloquine has antimalail effect while alone they have no significant effect.
Why ICR mice were used for this study ? Did the author also see drug response in Balb/c, C57Bl/6, or other strains of mice.
I have also concerns about survival data with NK65 strain. These mice are dying very quickly while other strains such as Swiss Webster and C57BL/6 survive with this strain for about 18-20 days.
The authors should provide microscopy images of infected parasites to see if the drugs induced death as suggested by the data.
Author Response
Thank you very much for your important comments and revision for improvement of our manuscript.
1-Why Nk65 mice are used. Authors should see the effect of drug on Anka strain which more closely mimic the human infection.
In our in vivo antimalarial study, we used the P. berghei NK65 strain in ICR mice for 20 years. We had many basic data for P. berghei NK 65 strain in ICR mice and the system is very familiar to our group. Specially, the system is easy to evaluate for new drug candidates in vivo because all control mice died due to parasites without host-concerned effect till 10 days. Many researchers use the ANKA strain in vivo as you mentioned but we don’t have the strain. We are thinking of changing the mouse type and parasite strain in the further experimental design, but it is necessary to accumulate basic research data, which cannot be applied immediately. In the future, we would like to consider setting up the evaluation system shown in the reviewer's comments.
2-I have concerns on how the parasitemia was detected. The parasitemia does not go reach25% at day 4 with this strain. I assume authors have added a very unusual high number of infected RBCs in the mice and that makes their result hard to interpret.
We apologize for the missed parts of Figures 3a. and 4a We checked the raw data and the figures are now correct. Detailed parasitemia on each day in Table 2 is correct. In Table 2, the parasitemia of the control group was 25.04 ± 3.3 and 33.20 ± 4.94 in Table 3. The data indicated that P. berghei NK65 strain infected ICR mice increased more than 25% of parasitemia in the control group on day 4 and the results are general. In the Methods part, we explained that the healthy ICR mice were inoculated with 1x106 P. berghei (NK 65 strain) infected erythrocytes by intravenously.
3-To my opinion based on results shown in Table, it is clear N89 and mefloquine has antimalail effect while alone they have no significant effect.
In the study, we designed a treatment schedule of 1 time td/day for 4 continuous days in N-89 and MQ alone. MQ data on days 4 and 9 showed significance versus N-89 in table 2, and days 7 and 9 in Table 3 using the statistical analysis. The result of a single drug is similar to your opinion. The above treatment schedule didn’t cure the effect of each compound. Based on the result of a single treatment drug, we planned combination therapy of N-89 and other antimalarials. As shown in Table 3 and Figure 4a, the combination of N-89 and MQ therapy increased the antimalarial and cure effect too. Additionally, end of December 2022, we reported the antimalarial effect of the transdermal N-89 in vivo (Aly et al., 2023. Parasitology International Journal [30]). Transdermal N-89 2 times/day for 4 consecutive days was required for the cure effect on P. berghei NK 65 strain in ICR mice. We added the above sentences in the discussion part of the revised MS too (page 10 lines 11-13).
4- Why ICR mice were used for this study ? Did the author also see drug response in Balb/c, C57Bl/6, or other strains of mice.
We used ICR mice for P. berghei NK65 strain maintaining and screening of antimalarial candidates in our laboratory for 20 years. We didn’t check other mice strains in our experiment, and we are thinking of changing the mouse strain in the next experimental design according to the reviewer’s comments.
5-I have also concerns about survival data with NK65 strain. These mice are dying very quickly while other strains such as Swiss Webster and C57BL/6 survive with this strain for about 18-20 days.
We had many basic data of P. berghei NK 65 strain in ICR mice for 20 years and the system is very familiar to our study group. Specially, the system is easy to evaluate for new drug candidates in vivo because all control mice died due to parasites without host-concerned effect till 10 days. In the MS, healthy mice were inoculated by iv infusion of 1x106 P. berghei-infected erythrocytes. The survival day of control mice was 6.6 ± 2.3 (low dose) and 6.4 ± 2.6 (high dose) in Table 4. We experienced many similar data of survival in ICR mice with P. berghei (NK 65 strain) and the result data was also usual even though your comments described the survival of 18 to 20 days in Swiss Webster and C57BL/6 mice. We are thinking of different mouse strain in the next experimental design according to the reviewer’s comments.
6-The authors should provide microscopy images of infected parasites to see if the drugs induced death as suggested by the data.
We used microscopy without photo media, and we are very sorry for didn’t provide the smear photos. We are planning the microscopy with photo system in the further study according to the reviewer’s comments.